# What leads to generalization of object proposals?

**Abstract.** Object proposal generation is often the first step in many detection models. It is lucrative to train a good proposal model, that generalizes to unseen classes. This could help scaling detection models to larger number of classes with fewer annotations. Motivated by this, we study how a detection model trained on a small set of source classes can provide proposals that *generalize* to unseen classes. We systematically study the properties of the dataset – visual diversity and label space granularity – required for good generalization. We show the trade-off between using fine-grained labels and coarse labels. We introduce the idea of prototypical classes: a set of sufficient and necessary classes required to train a detection model to obtain generalized proposals in a more data-efficient way. On the Open Images V4 dataset, we show that only 25% of the classes can be selected to form such a prototypical set. The resulting proposals from a model trained with these classes is only 4.3% worse than using all the classes, in terms of average recall (AR). We also demonstrate that Faster R-CNN model leads to better generalization of proposals compared to a single-stage network like RetinaNet.

**Keywords:** object proposals, object detection, generalization

## 1 Introduction

Object detection systems have shown considerable improvements for fully supervised settings [27, 18, 20, 26, 3], as well as weakly supervised settings [7, 1, 32] that only use image-level labels. Both approaches typically consider detection as a combination of two tasks: (a) spatial localization of the objects using proposals and (b) classification of the proposals into correct classes. A generalized proposal model that localizes all classes can help in scaling object detection. This could lead to the use of fewer or no bounding box annotations to only solve the classification task and development of more sophisticated classifiers, as explored in works like [35, 30].

Many detection models [27, 18] have been developed in recent years, which can be used to obtain high quality object proposals. However, an equally important aspect that determines the generalization ability of proposals is *the dataset* used to train these models. As illustrated in Fig. 1, the objects and class labels in a dataset significantly impact the ability to generalize to new classes. Intuitively, to localize a fine-grained vehicle like taxi in a target dataset, it might be

Fig. 1: Proposal models learned on seen vehicle classes can localize unseen classes which share similar localization structure like "bus" and "taxi". However, "barge" and "gondola", which are also vehicles will not be precisely localized by this model, due to lack of visual diversity in the training dataset for vehicles

sufficient to train a localization model with other vehicles like cars or vans in the source dataset. For localization (unlike classification), we may not need any training data for this class. On the other hand, training with these classes will not help in localizing other vehicles like boat.

While few works leverage this intuition for weakly supervised learning [35], the extent to which object localization depends on the categories used to train the model has not been well quantified and studied in detail. Towards this end, we define "generalization" as the ability of a model to localize (not classify) objects not annotated in the training dataset. In our work, we answer the question: *What kind of dataset is best suited to train a model that generalizes even to unseen object classes?*

We further study the ability of popular detection models like Faster R-CNN [27] and RetinaNet [18] to generate proposals that generalize to unseen classes. These networks are designed to improve the detection quality for the small set of seen classes in the training dataset. We carefully study these design choices and provide a way to obtain proposals that generalize to a larger set of unseen classes.

We answer several questions about dataset properties and modeling choices required for generalized proposals:

– **What are the properties of object classes to ensure generalization of proposals from a model?** First, we show that it is crucial to have visual diversity to obtain generalized proposals. We need examples of different vehicles like "car" and "boats", even if the examples are only labelled as "vehicle". Further, we hypothesize the existence of *prototypical classes* as a subset of leaf classes in a semantic hierarchy that are sufficient and necessary to construct a dataset to train a model for proposal generalization. We define new quantitative metrics to measure these properties for any set of classes and show that it is possible to construct a small prototypical set of object classes. This has positive implications for large taxonomies, since it is sufficient to annotate examples only for the prototypical classes.

– **Does the label-granularity of the dataset affect generalization? If so, what is the coarsest granularity that can be used?** Coarse-grained labels ("vehicles" instead of "taxis") are significantly less tedious to annotate and more accurate than fine-grained labels. Past works like RFCNN-3000 [30] argued that a single super class might be sufficient to obtain good proposals. However, we show that there is a trade-off between using very few

coarse classes and large-number of fine-grained classes, and a middle-ground approach leads to best generalization.

- **What are the *modeling* choices that are critical for leveraging state-of-the-art detectors to obtain generalized proposals?** We show that: (a) detections from two-stage networks like Faster R-CNN are better for obtaining generalized proposals than a single-stage network like RetinaNet, (b) while class-specific bounding box regression is typically used in Faster R-CNN, it is beneficial only when considering larger number of proposals (average recall AR@1000) and class-agnostic regression is better when considering fewer proposals (AR@100) and (c) choice of NMS threshold is dependent on the number of proposals being considered (AR@100 or AR@1000).

On OIV4 [16], we show that compared to training with all the object classes, using a prototypical subset of 25% of the object classes only leads to a drop of 4.3% in average recall (AR@100), while training with 50% of such classes leads to a negligible drop of 0.9%. We also show how the detections from Faster R-CNN can be fused to obtain high quality proposals that have 10% absolute gain in AR@100 compared to the class-agnostic proposals of the RPN from the same network and 3.5% better than RetinaNet. To stress the practical importance of generalized proposals, we also show that generalization ability is directly correlated with the performance of weakly supervised detection models.

## 2    Related Work

**Generalizing localization across multiple classes:** The idea of different object classes sharing the same structure has been exploited in building detection models for a long time[5, 22, 23, 29, 34]. More recently, [3, 27] also have a dedicated proposal network for object localization. However these works do not measure the transferability of proposals trained on one set of classes to another.

Uijlings *et al.* [35] tried to transfer information from coarse source classes to fine-grained target classes that share similar localization properties. They showed that this can help weakly supervised detection for the target classes. LSDA [11] transformed classifiers into detectors by sharing knowledge between classes. Multiple works [33, 12, 28, 9] showed the benefit of sharing localization information between similar classes to improve semi supervised and weakly supervised detection. Yang *et al.* [37] trained a large-scale detection model following similar principles. Singh *et al.* [30] showed that even a detector trained with one class can localize objects of different classes sufficiently well due to commonality between classes. We generalize this idea further. There has also been work on learning models [37, 26, 6] with a combination of bounding boxes for certain classes and only class labels for others. They inherently leverage the idea that localization can generalize across multiple classes. We provide systematic ways to quantify and measure this property for proposal models.

**Object proposal generation models:** There have been many seminal works on generating class-agnostic object proposals [36, 38, 25, 14]. A comprehensive

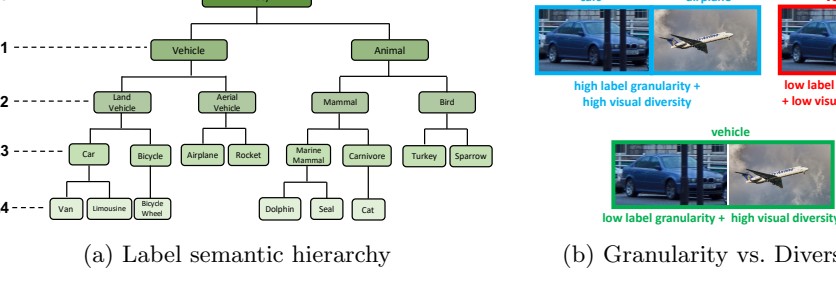

(a) Label semantic hierarchy          (b) Granularity vs. Diversity

Fig. 2: We study two important dataset properties needed to train a proposal model: label granularity and visual diversity. (a) Label granularity can be represented by different levels in a semantic hierarchy as shown. (b) The difference between label granularity and visual diversity is illustrated. At the same granularity, we can either have high or low visual diversity as shown

study of different methods can be found in [13] and a study of proposal evaluation metrics can be found in [2]. Proposal models have also been trained with dedicated architectures and objectives in [24, 15, 31]. In our work, we leverage standard models like Faster R-CNN and focus on the dataset properties required to achieve generalization with this model.

## 3  Approach

We study two important aspects involved in obtaining generalized proposals from a detection model:

(1) **Data Properties** such as the granularity of the label space (shown in Fig. 2a), and the visual diversity of object classes under each label, required for generalization of proposals. The idea of label granularity and visual diversity is shown in Fig. 2b. We investigate how a smaller subset of "prototypical" object classes in a dataset which is representative of all other classes can be identified.

(2) **Modeling Choice** for leveraging a detector trained on a dataset with seen classes to obtain proposals that generalize to unseen classes.

### 3.1  Dataset Properties

The choice of labels and data used to train the model is crucial for generalization. To study these properties, we assume: (a) classes are organized in a semantic tree and (b) internal nodes do not have any data of their own, that are not categorized into one of its child nodes. In practice, such a hierarchy is either already available (OIV4) or can be obtained from Wordnet [21]. These assumptions help us study the datasets under controlled settings. However, later we explore a way to identify "prototypical" subsets even when a semantic hierarchy is unavailable.

**Label Space Granularity** As we noted through some examples earlier, it is intuitive that we might not need fine-grained labels to train a good localization

model. To quantitatively study the effect of granularity, we construct different datasets with the same set of images and object bounding boxes, but consider classes at different levels of semantic hierarchy (Fig. 2a). We then train a model with these datasets and evaluate the generalization ability as a function of label granularity. For instance, for the coarsest root level, we assign all the bounding boxes the same "object" label and train a detector to distinguish objects from all non-objects. This pertains to the idea of objectness used in weakly supervised algorithms [36] and super-class in [30]. For an intermediate level, we collapse all leaf-labels to their corresponding parent labels at that level to train the model. While a fine-grained label space provides more information, a model trained at this level also attempts to distinguish object classes with similar structure and this could affect generalization. We quantify this trade-off in Sec. 4.3.

**Prototypical classes to capture visual diversity** One of the main aims of our work is to see if we can identify a significantly smaller number of classes than the full object-label space, so that bounding boxes from this set of classes are sufficient to train a generalized proposal model. Note that in Sec. 3.1, we wanted to study if a small set of coarse labels are sufficient to train a generalized proposal model. However, this does not answer anything about the visual diversity of objects within each sub-category that is required for generalization. As an example (shown in Fig. 2), in order to localize different types of vehicles like "car" or "airplane" it might be sufficient to collapse the label for all these objects into a single label named "vehicle", however dropping all instances of airplane during training will lead to a drop in performance for this class.

To quantitatively study this effect, we introduce the notion of "prototypical" classes. Given a large set of leaf classes, these are the smallest subset such that a model trained only with instances from them is sufficient to localize objects from the remaining classes. Note that due to the long-tail distribution of real-world data, obtaining images for large number of semantic classes is a tedious task. If a small set of prototypical classes does exist, this makes the data collection process much easier when scaling detection to large number of classes.

**Properties:** We identify the two properties that are required to quantify the prototypicality of a set of classes :

*Sufficient set*: is a set of classes such that training a model only with examples from them should be sufficient to localize objects from all other classes. The most superfluous sufficient set would be the entire set of leaf classes themselves.

*Necessary set*: is a set of classes such that dropping any class from this set will lead to a significant drop in generalization. A simple example would be a very coarse vertical like "vehicle". Intuitively dropping all vehicles would affect their localization as they do not share localization properties with other classes.

We provide concrete ways to measure both these properties in Sec. 4.3.

**Identifying prototypical classes:** Given a set of $N$ leaf classes $\mathbb{C}$, we wish to identify a set of $P$ prototypical classes $\mathbb{P} \subset \mathbb{C}$. Intuitively, this is similar to clustering the classes that have the same localization structure and then choosing a representative class from each cluster. Below, we discuss three approaches:

(a) **Oracle visual clustering**: To get an upper bound for choosing the best $P$ prototypical classes, we assume that bounding box annotations for all the $N$ leaf classes are available. We then use these bounding boxes to compute visual similarity between classes. We note that this is not a practical approach, but is crucial to evaluate the effectiveness of proxies we introduce later.

We first train a detection model using the annotations of all the leaf classes. We then measure the visual similarity between two classes $i, j$ as

$$S_{ij} = \max \left( \frac{AP^i(j)}{AP^j(j)}, \frac{AP^j(i)}{AP^i(i)} \right), \tag{1}$$

where $AP^i(j)$ is the detection average precision (AP) for the $j^{th}$ class when we use the detections corresponding to the $i^{th}$ class as detections of class $j$. $S_{ij}$ is a measure of how well one class can replace another class in localizing it. We then use the resulting similarity measure to hierarchically cluster the classes into $P$ clusters using agglomerative clustering. We then pick the class with the highest number of examples in each cluster to construct the set of prototypical classes. For practical reasons, we use frequency to choose the representative class, since this results in the construction of the largest dataset.

(b) **Semantic clustering based on frequency**: Semantic similarity is often viewed as a good proxy for visual similarity as shown through datasets like Imagenet [4] and OIV4. Hence, we use the semantic tree to cluster the classes in an hierarchical fashion starting from the leaves. At any given step, we cluster together two leaf classes that share a common parent if they jointly have the lowest number of examples. The algorithm stops when $P$ clusters are left. We then select the most frequent class from each cluster as a prototypical class. Here we assume that apriori we know the frequency of each class in a dataset. This is a very weak assumption, since a rough estimate of class distribution in a dataset can often be obtained even from weak labels like hashtags. This doesn't require any image-level label or bounding boxes and is easy to implement in practice.

(c) **Most frequent prototypical subset**: For this baseline, we choose the top $P$ most frequently occurring classes in the dataset as the prototypical classes. Note that unlike the previous approaches, this does not require any knowledge of the semantic hierarchy.

## 3.2   Modeling Choice

Once the dataset is fixed, the next step is to train a detection model. In our work, we explore the use of two models: Faster R-CNN and RetinaNet. The observations made in our work should nevertheless generalize to other two-stage and single-stage detection models as well.

In the case of a single-stage network, the detections from a model trained on a source dataset with seen classes can directly be treated as proposals. Their ability to localize novel classes in a target dataset can be evaluated to test generalization. However, for a two-stage network, another natural choice would be to use the Region Proposal Network (RPN) of the model, since it is trained in a class-agnostic fashion and aims to localize all objects in the image. However, as

noted by He et al. [10], the detection part of the model is better at localizing the object due to more fine-tuned bounding box regression and better background classification. We study this more rigorously, by comparing the generalization of proposals obtained from the detection head as well as RPN. We vary different model parameters to obtain the optimal setting for proposal generalization.

## 4    Experiments

We evaluate the ability of the object proposal obtained from detection models learned with different settings in Section 3.2 to generalize to new unseen classes. We also explore the effects of label-space granularity and the need for semantic and visual diversity. Finally, we show that a small set of prototypical classes could be used to train an effective proposal model for all classes in the dataset.

### 4.1    Experimental Setup

**Source and target splits:** We split each dataset into two parts: (a) *Source dataset* consisting of a set of seen classes called *source classes* and (b) *Target dataset* consisting of a set of unseen classes called *target classes*. *Target dataset* is used to evaluate the generalization of proposal models trained with the *Source dataset*. Since an image can contain both source and target classes, we ensure that such images are not present in the source class dataset. However, there may be a small number of images in the target dataset that contain source classes. We use the following two datasets for our experiments:

(1) *Open Images V4 (OIV4) [16]* consists of 600 classes. We retain only object classes which have more than 100 training images. This results in a total of 482 leaf classes. We randomly split all the leaf classes into 432 source (OIV4-source dataset) and 50 target (OIV4-target dataset) classes. There are also annotations associated only with internal nodes (for example, "animal") and without a specific leaf label (like the type of animal). We remove such annotations and all associated images, since such images cannot be unambiguously assigned to a source or target split. This leaves us with $1.2M$ images with $7.96M$ boxes in the train split and $73k$ images with $361K$ boxes in the test split. For training proposal models, we always use the train split and for evaluation we use the test split. Wherever needed, we explicitly suffix the dataset with "train" and "test" (for example, OIV4-source-train and OIV4-source-test).

(2) *COCO [19]*: We use the 2017 version of the COCO dataset and randomly split the classes in to 70 source (COCO-source dataset) and 10 target (COCO-target dataset) classes. For training, we use the train split and for evaluation, we use the 5000 images from the validation set. Wherever needed, we explicitly suffix the dataset with "train" and "test".

Target classes list is provided in the supplementary.

**Evaluation metrics:** We report the standard average recall (AR@k) [13] metric to evaluate the quality of proposals. One of the main motivations for building a generalized proposal model is to use the resulting proposals to train detection

models for unseen classes with limited or no bounding box annotation. A typical proposal-based supervised detection model RCNN could also be used to evaluate the quality of proposals. However, the application to weakly supervised detection is more compelling since their performance is closely tied to proposals than supervised models which can correct the inaccuracies in proposals due to availability of labelled bounding boxes. Hence, we implement a weakly supervised detector with the approach used in YOLO9000 [26][1]. We report the detection AP (averaged over IoU thresholds ranging from 0.5 to 0.95) on the test set of the target dataset. Please see the supplementary material for more details.

**Implementation details:** We fix Imagenet pre-trained ResNet-50 with Feature Pyramid Networks [17] as the backbone for all models. We use the Detectron codebase [8]. For COCO, we train the models for $90k$ iterations with an initial learning rate and the decay suggested in [27]. For OIV4, we train the models for $800k$ iterations with an initial learning rate of 0.01 and cosine learning rate decay. When training the weakly supervised model ([26]), we use the top 100 proposals in each image to choose pseudo ground truth at every training iteration.

## 4.2   Modeling Choices

We first identify the best detection model and setting to extract proposals that generalize to new unseen classes. We then analyze generalization ability under different settings from this model. We reiterate that in order to test generalization, evaluation is done on target classes that have no intersection with the source classes used during training.

**Choice of detection model:** We compare the generalization ability of a two-stage network (Faster R-CNN) and a single-stage network (RetinaNet) in Fig. 3a. Since, in a two-stage model like Faster R-CNN, the output from the RPN is class-agnostic and can be used as proposals too, we compare the performance of the RPN as well. The models are trained on COCO-source-train dataset. We report AR@100 on seen classes in the COCO-source-test dataset, as well as unseen classes in the COCO-target-test. The difference in performance between seen and unseen classes reflects the generalization gap. We also show an upper-bound performance on COCO-target-test obtained by models trained on the full training dataset containing both COCO-source-train and COCO-target-train.

We notice that on seen classes, RetinaNet achieves a lower performance compared to Faster R-CNN (drop of 2.4%). However, the drop is larger for unseen target classes (3.5%), indicating a larger generalization gap for RetinaNet. One reason for this is that RetinaNet is more sensitive to missing bounding boxes corresponding to unlabelled unseen classes in the source dataset. Proposals corresponding to unseen object classes that are not annotated in the training data are treated as hard-negatives, due to the use of focal-loss. Hence, the model heavily penalizes proposals corresponding to unannotated bounding boxes, leading to overall drop in AR. Since some seen classes share visual similarity with

---

[1] We chose [26] due to its simplicity. In practice, we can use other weakly supervised approaches too.

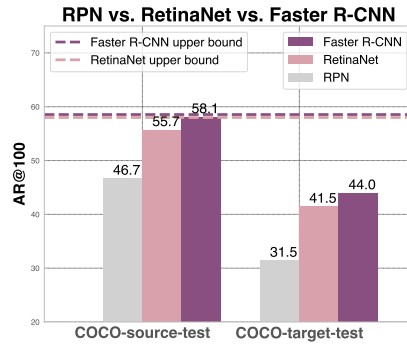
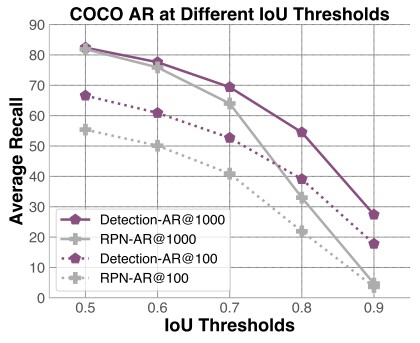

(a) Comparison of detection models        (b) RPN vs. detection head

Fig. 3: (a) AR@100 corresponding to different models trained on COCO-source-train and evaluated on different test splits. Upper-bound corresponds to model trained on full COCO dataset and evaluated on COCO-target-test. (b) Average recall of RPN and detection head at different IoU thresholds, for model trained on COCO-source-train and evaluated on COCO-target-test

unseen classes, this sensitivity to missing annotations affects AR for seen classes too. However, this effect is more magnified for unseen target classes. On the other hand, in Faster R-CNN, only a small number of proposals (less than 512) which do not intersect with annotated bounding boxes are sampled at random as negatives. The probability that a proposal corresponding to an unseen object class is chosen as a negative is lower, leading to better generalization. Hence, for the rest of the paper, we use Faster R-CNN as the detection model.

We also notice that the detection head of Faster R-CNN provides better overall performance *without* sacrificing generalization. This can be attributed to better bounding box regression from the detection head which has additional layers, following the RPN in the model. To investigate this effect, we measure AR at different IoU thresholds for both sets of proposals for the model trained on COCO-source and evaluated on COCO-target in Fig. 3b. We see that the difference in AR@1000 increases drastically at higher values of IoU threshold, and is negligible at a threshold of 0.5. This implies that the boxes from the detection head are more fine-tuned to exactly localize objects, unlike the RPN.

**Choice of Faster R-CNN settings:** The results so far were obtained using class-specific bounding box regression (which is the standard setting in Faster R-CNN) for the detection head. Since we want the bounding boxes to generalize to unseen classes, class agnostic regression could be a valid choice too. We study this in Fig. 4 for OIV4 and COCO. We see that class agnostic regression is better for small number of proposals as seen by AR@10,20,50. However, when we consider more proposals (AR@1000), class specific regression provides a significant gain (4.5% for OIV4 and 7.5% for COCO). It results in multiple regressed versions (one corresponding to each class) of the same proposal generated from the RPN. This helps in improving recall at higher number of proposals.

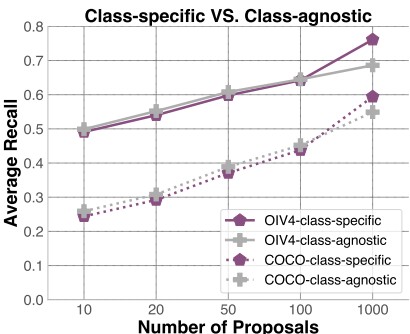

Fig. 4: Effect of class agnostic regression vs. class specific regression

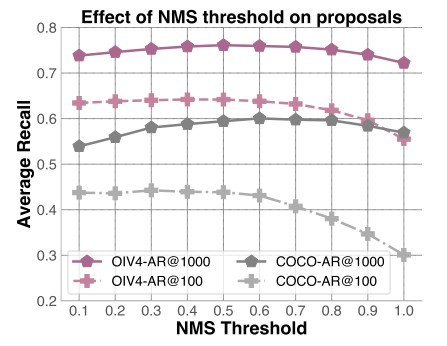

Fig. 5: Effect of NMS threshold on performance of proposals

Previously, we fixed the NMS threshold to 0.5. We study the effect of this threshold in Fig. 5. We train on OIV4-source, COCO-source and test on OIV4-target, COCO-target respectively. Intuitively, a low threshold can improve spatial coverage of objects by ensuring proposals are spatially well spread out. When considering a larger number of proposals, there are sufficient boxes to ensure spatial coverage, and having some redundancy is helpful. This is witnessed by the steeper drop in AR@1000 at low NMS thresholds, unlike AR@100.

Based on these observations, we use class-specific bounding box regression with an NMS threshold of 0.5 for rest of the experiments.

Table 1: Comparing performance of proposals generated by RPN head and detection head for weakly supervised detection. We also show the AR@100 numbers which are seen to be correlated with detection AP

| | Target Dataset - OIV4-target | | | |
| | Source: OIV4-source | | Source: OIV4-all | |
| | Det. AP | AR@100 | Det. AP | AR@100 |
|---|---|---|---|---|
| Faster R-CNN RPN | 8.7 | 55.0 | 9.6 | 60.4 |
| Faster R-CNN Detection | **24.0** | **69.4** | **30.8** | **76.9** |

**Weakly supervised detection:** A strong practical utility for generalized proposals that localize all objects is that, no bounding box annotations should be needed to train a detection model for new object classes. Hence, we measure the effect of better generalized proposals on the performance of a weakly supervised detection model, trained without bounding box annotations. We show results corresponding to the RPN head and detection head of Faster R-CNN in Tab. 1. The weakly supervised model is trained on OIV4-target-train and evaluated on OIV4-target-test. We also show results for proposals obtained from training with OIV4-source as well as OIV4-all (upper-bound). We see that the performance of the weakly supervised detection model is directly correlated with the quality of the proposals being used, showing the need for good generalized proposals.

## 4.3    Dataset Properties

**Effect of label space granularity:** OIV4 organizes object classes in a semantic hierarchy with 5 levels. We directly leverage this hierarchy to measure the effect of label granularity (Fig. 2a). We construct a dataset at each level $L_i$ (OIV4-source-$L_i$) by retaining all the images in OIV4-source, but relabeling bounding boxes corresponding to leaf labels with their ancestor at $L_i$. We construct 5 datasets, one for each level with the same set of images and bounding boxes.

We report the performance of these models on OIV4-target in Tab. 2. Along with AR@100/1000, we also report the detection AP of the weakly supervised detection models trained with the proposals obtained from the corresponding levels. The weakly supervised models are trained on OIV4-target-train and evaluated on OIV4-target-test.

Table 2: Effect of different label space granularities on the quality of proposal for OIV4 dataset. The number of classes at each level is shown in brackets. Evaluation is done on OIV4-target-eval dataset. Both AR and weakly supervised detection AP are reported

| Source Dataset | AR@100 | AR@1000 | AP (weak) |
|---|---|---|---|
| OIV4-source-$L_0(1)$ | 61.7 | 72.0 | 19.5 |
| OIV4-source-$L_1(86)$ | 63.4 | 73.0 | 22.6 |
| OIV4-source-$L_2(270)$ | 63.7 | 75.2 | 23.1 |
| OIV4-source-$L_3(398)$ | 65.2 | 77.2 | 24.3 |
| OIV4-source-$L_4(432)$ | 64.2 | 76.1 | 24.0 |

Some past works like [30] postulated that one super-class (similar to $L_0$) could be sufficient. However, we observe that both AR@100 and AR@1000 increase as we move from $L_0$ to $L_1$ along with a significant gain (3.1%) in AP. This indicates that training with just a binary label yields lower quality proposals compared to training with at least a coarse set of labels at $L_1$. While both AP and AR@100 increase as the granularity increases from $L_1$ to $L_3$, the difference is fairly small for both metrics ($< 2\%$ change). However, annotating bounding boxes with labels at $L_1$ (86 labels) is significantly cheaper than $L_3$ (398 labels). Hence, $L_1$ can be seen as a good trade-off in terms of labelling cost, and training a good model.

**Need for visual and semantic diversity:** We noticed that training with coarse labels can yield good proposals. It would be interesting to observe if all or only some of these coarse classes are crucial to build a good proposal model. To study this, we conduct ablation experiments where we train a model with OIV4-source-train after dropping all images having a specific $L_1$ label and evaluate the proposals on the OIV4-source-test images belonging to this label in Fig. 6a. We repeat this experiment for a few fine-grained classes at $L_4$ in Fig. 6b.

We notice that certain coarse classes (like "clothing" and "vehicle") experience a huge drop in performance. On the other hand, "animal" and "food" are less affected. This can be explained from the fact that, there are many toy-animal images within the coarse label "toy", similarly "containers" is a coarse class in OIV4 which is often depicted with food in it. These classes can act as

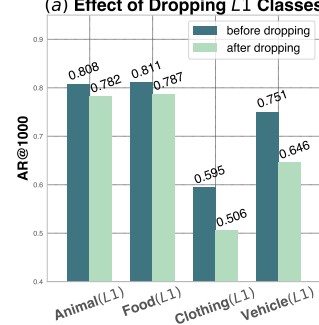 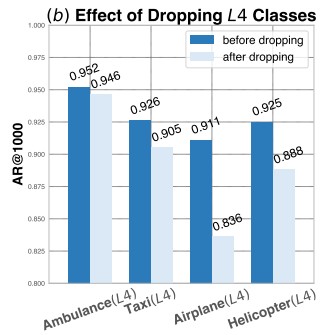

Fig. 6: Effect of Semantic Diversity, measured by dropping an object class during training and measuring the resulting change in AR for that class: (a) dropping L1 classes and (b) dropping L4 classes

proxies for "animal" and "food" respectively. However, "clothing" and "vehicle" do not have good proxies. More interestingly, we make a similar observation for finer classes at $L_4$ like airplanes and helicopters. This suggests that there is a smaller set of objects that have unique localization properties in OIV4.

**Prototypical classes:**  Some object classes are similar to others in terms of localization, while there are classes that are unique and need to be included in training. Motivated by this observation, we try to identify a small set of classes called "prototypical" classes which are both necessary and sufficient to train a generalizable proposal model.

We use the OIV4-source dataset as before with 432 leaf classes. We use the different approaches outlined in Sec. 3.1 to identify a subset of "prototypical" classes. Note that among these methods, oracle visual clustering assumes availability of bounding boxes for all classes and serves as an upper bound on how to identify a really good prototypical set. Some sample clusters of classes obtained by this method are shown in Tab. 3. The remaining methods make weaker assumptions and are more useful in practice. In addition to these methods,we also train models with a set of randomly chosen prototypical classes.

Table 3: Sample clusters obtained by oracle visual clustering for $P = 50$. The most frequent class in each cluster chosen as a prototypical class is highlighted

| **Woman**, Girl, Doll | **Wheel**, Tire, Bicyclewheel | **Lobster**, Scorpion, Centipede |
|---|---|---|
| **Glasses**, Goggles | **Jeans**, Shorts, Miniskirt | **Goose**, Ostrich, Turkey |
| **Book**, Shelf, Bookcase | **Musicalkeyboard**, Piano | **Swimmingpool**, Bathtub, Jacuzzi |
| **Man**, Boy, Shirt | **Apple**, Pomegranate, Peach | **Raven**, Woodpecker, Bluejay |

We introduce two ways to measure *sufficiency* and *necessity*. From the 432 classes, once we pick a subset of $P$ prototypical classes, we train a proposal model and evaluate the resulting model on the 50 target classes in OIV4-target, to measure *sufficiency* and *necessity*.

**Dataset construction for fair comparison** We ensure that the total number of images as well as bounding box annotations are kept fixed when we construct datasets for different prototypical subsets. This is important to ensure that pro-

posals trained with different subsets are comparable. Once we chose a set of $P$ prototypical classes, we uniformly sub-sample OIV4-source images having any of these prototypical classes to get a subset of $920K$ images. And within each subset, we uniformly sub-sample the bounding boxes corresponding to the prototypical classes to retain $5.2M$ bounding boxes. We do not retain any bounding boxes outside the chosen prototypical classes.

**Training with prototypical subsets** For a set of prototypical classes and the corresponding dataset, we train a Faster R-CNN with those classes as labels. We combine the detections as described in Sec. 3.2 to obtain proposals.

**Measuring sufficiency of prototypical classes** A subset of classes are sufficient, if a proposal model trained with them generalizes as well as a model trained with all classes. We follow this notion and evaluate the proposals obtained from the models trained with different prototypical subsets on OIV4-target and report the average recall (AR@100) in Fig. 7a. Similar trends are observed with AR@1000 as well (shown in supplementary).

Looking at the proposals obtained from oracle visual clustering, training with less than 25% of the classes (100) leads to only a drop of 4.8% in AR@100, compared to training with images belonging to all object classes. This gap reduces to 0.4% if we train with 50% (200) of all the classes. This provides an empirical proof for the existence of a significantly smaller number of object classes that are sufficient to train a generalizable proposal model.

Next, we look at the prototypical classes obtained from a more practical approach: semantic clustering. We notice that the proposal model trained with these prototypical classes always outperform other approaches such as choosing a random set of classes or the most frequent set of classes. Further, the performance of this method is only lower by a margin of 3% compared to oracle visual clustering for different value of $P$. Selecting most frequent set of classes as the prototypical subset performs slightly worse than semantic clustering. This shows that semantic clustering can serve as a good way to identify prototypical classes for large taxonomies when the semantic hierarchy is available for the dataset, else the most frequent subset is a weaker alternative.

**Measuring necessity of prototypical classes** A set of classes are considered necessary, if there is no redundancy among the classes in terms of localization properties. For a given class in the set, there should be no equivalent class which can provide similar bounding boxes. We measure this property for a prototypical subset by evaluating the corresponding proposal model on OIV4-target dataset using the following method. For every target class in OIV4-target, we measure the relative change in AR@100 and AR@1000 by removing proposals corresponding to the most similar class in the prototypical subset (similarity measured by Eq. 1). The change in AR would be minimal if there is another class in the prototypical subset which can localize the target class. This measure, averaged over all target classes provides a good estimate of necessity. A high value symbolizes a high degree of necessity, while a low value corresponds to redundancy among the prototypical classes. We plot this for different number of prototypical classes for oracle visual clustering and semantic clustering in Fig. 7b.

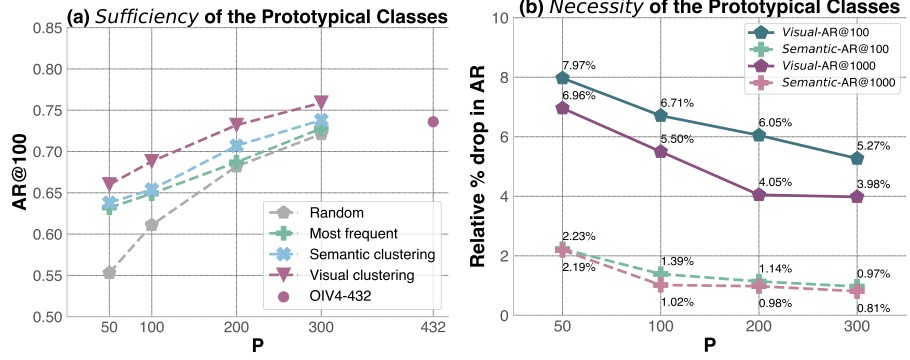

Fig. 7: (a) Average recall AR@100 for proposals obtained from models trained with varying number of prototypical classes chosen by different methods. We show the average recall on the OIV4-target dataset with 50 unseen classes. $P$ denotes the number of prototypical classes. Higher value indicates higher sufficiency. (b) The relative change in AR for target classes by dropping proposals corresponding to the most similar class in the prototypical subset. Higher value indicates lower redundancy in prototypical subset and higher necessity

We notice that at any given number of prototypical classes, the change in average recall is higher for oracle visual clustering compared to semantic clustering. This demonstrates that visual clustering leads to prototypical classes which are less redundant (and more necessary). As expected, we see the necessity drops, as we increase the number of prototypical classes for both methods. Again, this is expected since redundancy between classes increases with more number of classes. The relative change in AR@1000 is also seen to be lower than AR@100, since when considering a larger number of proposals, we expect more redundancy among the proposals. Finally, for oracle visual clustering as we move from 200 to 300 classes, sufficiency changes by a small amount from 73.2 to 75.9 ( Fig. 7a), while the necessity drops steeply in Fig. 7b. This suggests that the ideal number of prototypical classes for OIV4 could be around 200.

## 5    Conclusion

We studied the ability of detection models trained on a set of seen classes to localize unseen classes. We showed that Faster R-CNN can be used to obtain better proposals for unseen classes than RetinaNet, and studied the effect of model choices on generalization of proposals, like class-agnostic bounding box regression and NMS threshold. We quantitatively measured the importance of visual diversity and showed that using a very fine-grained or very coarse labelspace can both affect generalization, while a middle-ground approach is best suited. We introduced the idea of prototypical classes that are sufficient and necessary to obtain generalized proposals. We demonstrated different approaches to determine small prototypical subsets for a given dataset. We believe that our work is a step forward towards learning proposals that generalize to a large number of classes and scaling up detection in a more data-efficient way.

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
