# OpenReview forum: "What leads to generalization of object proposals?"
_thecvf.com/ECCV/2020/Workshop/VIPriors — VIPriors Poster_

### Official Review · AnonReviewer1 · 2020-07-21
**Prototype classes can be sufficient to localize in object detection**

**Confidence:** 3
**Rating:** 7

**Review:**

[Summary] In 2-3 sentences, describe the key ideas, experiments, and their significance.

The authors propose to learn object localizations using only prototype classes. They explore what defines prototype classes and experiment with many ablations and hyperparameters to their method.

[Strengths] What are the strengths of the paper? Clearly explain why these aspects of the paper are valuable.

Powerful idea; clear definitions; practical modeling choices for determining prototypical classes; extensive experimentation.

[Weaknesses] What are the weaknesses of the paper? Clearly explain why these aspects of the paper are weak.

Marginally related to visual inductive priors.

[Overall rating] Paper rating: Accept

---

### Official Review · AnonReviewer2 · 2020-07-28
**What leads to generalization of object proposals?**

**Confidence:** 3
**Rating:** 7

**Review:**

1. [Summary] In 2-3 sentences, describe the key ideas, experiments, and their significance.

 In this paper, authors propose guidelines to build proper datasets for object proposals that can offer good generalization when training models on them. Concretely, the paper introduces the idea of prototypical classes as the sufficient and necessary classes to achieve good generalization. To proof this, they conduct a series of experiments on OIV4 and COCO datasets. As oracles, they choose Faster RCNN and RetinaNet.

2. [Strengths] What are the strengths of the paper? Clearly explain why these aspects of the paper are valuable.

 -	The paper is very well written. Story is very easy to follow.
 -	Generalization typically has been study from the model perspective. However, interpreting the problem from the data perspective is very interesting.
 -	Although authors focus on data, they also offer a study of what is happening with the models to validate the results.
 -	In particular, prototypical classes seems pretty interesting.

3. [Weaknesses] What are the weaknesses of the paper? Clearly explain why these aspects of the paper are weak.

 -	The effect of space granularity has a weird interpretation. The model used for this experiment is class specific Faster RCNN. Have the authors tried the class agnostic version?
 -	Visual and Semantic diversity seems obvious. It would be interesting to study also the amount of samples, as well as how similar they are.


4. [Overall rating] Paper rating.

 7

5. [Justification of rating] Please explain how the strengths and weaknesses aforementioned were weighed in for the rating.

 The whole paper is interesting. Even more from the efficiency perspective. Prototype classes play an important role.

6. [Detailed comments] Additional comments regarding the paper (e.g. typos or other possible improvements you would like to see for the camera-ready version of the paper, if any.)

---

### Decision · Program_Chairs · 2020-07-29

**Decision:**

Accept (Poster)

**Comment:**

It is our pleasure to inform you that your paper has been accepted to the poster track of 1st Visual Inductive Priors for Data-Efficient Deep Learning Workshop.

Please note the following deadlines:
* August 11, 2020 - workshop material, including:
 * paper in PDF format;
 * pre-recorded video presentation;
 * slides of the presentation in PDF.
* September 15, 2020 - camera-ready paper

The reviews can be found on OpenReview. Please take these comments and suggestions into account when preparing the camera-ready version of your paper, which is due September 15, 2020. The camera-ready paper should be uploaded to OpenReview.

As part of the workshop, each accepted paper must submit a pre-recorded 90 second talk before August 11, 2020. You will receive more information on how to upload the material shortly. The requirements for the video are:
* Duration: maximum 90 seconds
* MP4 format
* File size max. 100 MB
* Has an inset with a video of the speaker
* 16:9 aspect ratio (strongly preferred)
* 1920x1080 resolution (strongly preferred, at least 720 height)

Our suggested software for pre-recording your presentation is Zoom. For more information, please refer to the following guides:
How to record with Zoom Guide: http://homepages.inf.ed.ac.uk/rbf/ECCV2020HowtoRecordusingZoom.pdf
How to Record with Zoom tutorial: https://www.youtube.com/watch?v=CR199W7HdC0
Please ensure that at least one of the authors of the paper is available to attend the workshop during the allotted times. Note that the workshop will take place in two sessions spread across time zones (details are to follow). We will send instructions on how to connect to the workshop as soon as possible. The schedule for all talks and papers will be posted soon at the workshop website: https://vipriors.github.io.

We look forward to seeing you at the workshop!